# Waste generation and carbon emissions of a hospital kitchen in the US: Potential for waste diversion and carbon reductions

**Cassandra L. Thiel**[1,2]*, **SiWoon Park**[1], **Aviva A. Musicus**[3], **Jenna Agins**[4], **Jocelyn Gan**[4], **Jeffrey Held**[4], **Amy Horrocks**[4], **Marie A. Bragg**[1,5]

**1** Department of Population Health, NYU Grossman School of Medicine, New York City, New York, United States of America, **2** Department of Civil and Urban Engineering, NYU Tandon School of Engineering, New York City, New York, United States of America, **3** Department of Social and Behavioral Sciences, Harvard T. H. Chan School of Public Health, Boston, Massachusetts, United States of America, **4** NYU Langone Health, New York City, New York, United States of America, **5** Department of Nutrition, NYU School of Global Public Health, New York City, New York, United States of America

* cassandra.thiel@nyulangone.org

**Data Availability Statement:** The data underlying the results presented in the study are available in the supplemental file.

## Abstract

This study measured the total quantity and composition of waste generated in a large, New York City (NYC) hospital kitchen over a one-day period to assess the impact of potential waste diversion strategies in potential weight of waste diverted from landfill and reduction in greenhouse gas (GHG) emissions. During the one-day audit, the hospital kitchen generated 1515.15 kg (1.7 US tons) of solid waste daily or 0.23 kg of total waste per meal served. Extrapolating to all meals served in 2019, the hospital kitchen generates over 442,067 kg (487 US tons) of waste and emits approximately 294,466 kg of $CO_2e$ annually from waste disposal. Most of this waste (85%, 376,247 kg or 415 US tons annually) is currently sent to landfill. With feasible changes, including increased recycling and moderate composting, this hospital could reduce landfilled waste by 205,245 kg (226 US tons, or 55% reduction) and reduce GHG emissions by 189,025 kg $CO_2e$ (64% reduction). Given NYC's ambitious waste and GHG emission reduction targets outlined in its OneNYC strategic plan, studies analyzing composition, emissions, and waste diversion potential of large institutions can be valuable in achieving city sustainability goals.

## Introduction

In 2017, the United States produced over 41 million US tons of total food waste, of which 75% were landfilled, 19% were combusted, and only 6% were composted [1]. This high proportion of landfilled food waste produces negative environmental impacts that fuel climate change. Decomposing food in landfills generates methane, a potent greenhouse gas (GHG) that has a global warming potential (GWP) 104 times greater than that of carbon dioxide [2]. U.S. landfills contribute 95.6 billion kg of GHGs annually as of 2014 [3, 4]. Diverting food waste from landfills is a promising strategy to combat climate change.

**Funding:** Funding support was provided by an NIH Early Independence Award (5DP5OD021373-05) for MB (https://commonfund.nih.gov/earlyindependence). AAM is supported by NIH grant number 2T32CA057711-27. The content of this article is solely the responsibility of the authors and does not necessarily represent the official views of the NIH. The funders had no role in study design, data collection and analysis, decision to publish, or preparation of the manuscript.

**Competing interests:** The authors have declared that no competing interests exist.

Institutional change is crucial for waste diversion, and it can be spurred by citywide policies and guidance. For example, New York City (NYC) has recognized the threat of climate change to its communities and has set aggressive goals to reduce its environmental impact. Launched in April 2015, the city's OneNYC 2050 strategic plan established climate goals including zero waste to landfill by 2030 and achieving carbon neutrality by 2050 [5]. The city's strategic plan recognizes the important role that institutions can play in reducing the city's environmental impact. In February 2016, NYC launched the Mayor's Zero Waste Challenge (ZWC), which invited NYC businesses to divert at least 50% of their waste from landfill and incineration [6]. When ZWC ended in June 2016, the majority of businesses had reduced landfill waste production by at least 50%. Among the 39 participating locations, 12 locations diverted more than 75% of their waste and two diverted more than 90% of waste. ZWC was successful because it targeted businesses with high volumes of food waste and provided a platform for leadership recognition [6]. Since 2016, New York City has also passed multiple laws requiring large generators of food waste (e.g., arenas, chain restaurants, grocery stores) to put their wasted food to good use, such as donating excess edible food and composting food scraps [7, 8].

Despite NYC's recognition of the importance of institutional change to protect the environment, the organic waste management of city hospitals have not been included in these regulations. This is a potential missed opportunity, as NYC hospitals have been estimated to generate 5% of commercial food wastes in the city [9]. Most research on hospital food waste has focused exclusively on wasted food from patient trays, finding that a median of 30% of food on trays is thrown away uneaten [10–13]. National data suggest that hospitals produce 30 pounds of food waste per patient bed per day [14]. Few studies, however, have examined the volume, composition, and carbon footprint of broader food waste production at hospitals, which are crucial metrics to optimize and reduce food waste at the institutional level and could produce economic and environmental benefits [15]. To address the gap in the literature and inform potential institutional and regulatory actions, this study aims to analyze waste generation over one day in the kitchen of an academic medical center, including characterizing the waste for diversion potential and quantifying GHG emissions associated with its disposal. The auditing methodology utilized in this paper can serve as a model for other hospitals seeking to reduce their carbon footprint by diverting food waste from landfill.

## Methods

### Case location

We measured all waste generated over the course of a single day by New York University Langone Health's (NYU Langone) Main Campus on the east side of Manhattan. The Main Campus is a large academic medical center comprised of 10 interconnected acute care clinical, research and non-clinical buildings. The Main Campus has a total of 750 total beds, roughly 9,575 onsite Full Time Employees, and 212,574 patient-days annually. The NYU Langone Department of Food & Nutrition Services provides over 1,900 inpatient meals per day for several areas of the Main Campus—Tisch Hospital, the Kimmel Pavilion, and the Schwartz Health Care Center—and approximately 3,400 retail meals daily at the Tisch Café, Kimmel Café & Coffee Bar, Science Café & Coffee Bar, Café 41, ACC Café, and Orthopedic Center Café. This institutional scratch kitchen cooks fresh meals daily and develops menus based on seasonality and availability of fresh produce. Annually, the kitchen prepares over 1.9 million patient and retail meals.

The kitchen is divided into six workspaces or zones: 1) catering, which serves executive meetings and on-site conferences; 2) production, which includes main food preparation areas,

walk-in coolers and freezers; 3) patient services, which prepares trays for inpatient meals; 4) the pot room, where large dishes are washed; 5) the dish room, for all other dish and tray washing; and 6) the decanting room, where boxed food deliveries are received, unpacked, and distributed. Prior to the audit, the kitchen diverted cardboard and metal food packaging to a recycling waste stream, and all other wastes were sent to landfill. Ethics approval was not needed, as this was not human-subjects research.

## Data collection, waste auditing protocol

To measure and characterize all waste produced by the kitchen, we conducted a waste audit over the course of a single, typical (non-holiday) weekday in December 2019. In preparation for the audit, the research team met with food & nutrition services management, kitchen, and dietetic staff. The initial meeting included a tour of the kitchen and identification of high-level goals of the project, namely providing baseline data characterizing current waste outputs to evaluate the feasibility of a composting program. Excluded from the waste audit were the wastes generated from retail meals (that is, the waste generated by customers in the cafeterias) as these wastes are sent directly from the cafeteria to the hospital's waste disposal area. Recyclable materials disposed to landfill were recorded.

A second meeting set the details of the audit, including required physical resources (labeled trash bags, personal protective equipment for the auditors, and sorting tools such as scales, buckets, and recording devices) and staff protocols (notifications and potential training required prior to the waste sorting). Prior to the audit, we created masking tape labels for all 6 kitchen zones and affixed them on over 200 empty trash bags to ensure we noted origins of waste within the kitchen. Pathways and designated weighing sites were identified to ensure no waste was missed during the audit. All waste management and food service staff were informed of the date of the audit.

On the day of the audit, we utilized two different methods of data collection: (1) "all-day audit": generic weighing of every bag of waste leaving the kitchen during operating hours (5:30am–10:00pm); and (2) "detailed audits": three, one-hour periods where the contents of every bag of waste was sorted and weighed. For both methods of data collection, full bags of waste were weighed and characterized by the kitchen zones from which the bags originated. The time each bag reached the loading dock was also noted. We used two types of scales to weigh bags of waste: a Polysun portable hanging scale for lighter bags (bags that could easily be lifted out of trash cans), with an accuracy of 10 kg x 5 g or 10–50 kg x 10 g, and a Mettler Toledo IND560 floor scale in the loading docks for heavier bags or bins.

The detailed audits were performed during three specific time periods assumed to be times with higher waste-generation rates, based on meal service times: 9:30am to 10:30am, 12:00pm to 1:00pm, and 3:30pm to 4:30pm. All waste generated during these time periods was first weighed as full bags. We then opened and sorted each bag of waste into various categories and weighed each waste category separately. Waste was separated into the following categories: compostable food/organic wastes; recyclable glass, metal, beverage cartons and plastic; recyclable papers & paperboards; non-recyclable plastics; gloves used by food service workers; and other general wastes (including disposable flatware). We also made note of unused items, such as unopened drink cartons returned on patient trays. Here, we used an Edlund ERS-60 accurate to 30 kg x 5 g for weighing. The auditors rotated throughout the day, using the same data collection sheets. There were ten auditors in total, two of which served as supervisors to ensure consistency in data collection.

## Data analysis

Descriptive statistics were used to characterize the quantity and timing of waste generation from the all-day and detailed audits. We further estimated the composition of waste generated per day by calculating the proportion of waste in each composition category described previously (e.g., compostable food/organic wastes) from each kitchen zone during the detailed audits and multiplying this by the total daily waste from each kitchen zone. Annual waste generation was estimated by dividing daily results by the number of meals served on the audit day and multiplying by the total number of meals served in 2019 (696,485 inpatient meals and 1,248,410 estimated retail meals).

Based on waste composition estimates, we calculate three waste diversion scenarios: 1) the number of pounds of annual waste produced from existing paper and metal recycling with all other waste sent to landfill (current hospital practice or Base Case), 2) the number of pounds of annual waste in the ideal scenario in which everything that could be recycled or composted were diverted from landfill (ideal), and 3) the number of pounds of annual waste in the most-likely scenario of increased recycling and compost diversion based on the kitchen's setup and the likelihood of regular sorting occurring (likely future). In the likely future scenario, all dish room waste would go to landfill because the kitchen staff deemed sorting of dish room waste to be almost impossible due to staffing resources, safety, and space considerations, but all other recycling and compostable materials would be diverted from landfill. Of note, these scenarios do not include post-consumer waste from the cafeterias or retail meals.

Finally, we estimated annual GHG emissions for disposal and treatment of waste in each of the diversion scenarios described above. The conversion factor of kg carbon dioxide equivalents ($CO_2$e) were taken from the US EPA's Documentation for Greenhouse Gas Emission and Energy Factors Used in the Waste Reduction Model (WARM), Exhibit 5–1 (composting food waste) and Exhibit 7–16 (landfilling food waste) [16]. In this model, composted material produces a net negative GHG emission factor because compost acts as a carbon sink. We did not estimate GHGs from upstream activities (such as farming or cooking), and we did not issue any emissions or $CO_2$e credits to the hospital for the material it recycled, as it is standard practice within life cycle assessment methods to assign $CO_2$e reductions credit only to account for *utilized* recycled materials.

## Results

On the day of our audit, the total waste (trash, recyclables, and compostable waste) recorded leaving the kitchen weighed 1515.2 kg (1.7 US tons) in 171 bags. The average waste generation rate was 94.7 kg/hr (209 lbs/hr) and 10.7 bags/hr over the 16-hour audit. On this day, the hospital served 2,010 inpatient meals and 4,656 retail meals generating 0.23 kg (0.51 lbs) of total waste per meal. Assuming this represents an average daily waste generation, this case study kitchen will produce an average total waste of 442,067 kg (487 US tons or 974,590 lbs) per year and use approximately 49,892 plastic garbage bags.

Out of all kitchen zones, a majority of waste originated in the dish room (535.8 kg or 35% of total waste) where patient trays are cleaned and the pot room (309.8 kg or 20%) where cookware are cleaned (Fig 1). The next largest zones of waste generation were the production area for general food preparation (208.1 kg or 14%) and patient services, where inpatient trays are prepared (191.9 kg or 13%). The decanting room, where food deliveries are unpacked, generated 188.2 kg or 12% of total waste, most of which was cardboard boxes used for packaging. A small portion of this, by weight, was plastic wrap used to secure the boxes, but this was not weighed separately. The catering area generated only 54.6 kg or 4% of total waste. The remaining 26.8 kg (2%) of general waste came from four unlabeled bags thought to originate in the

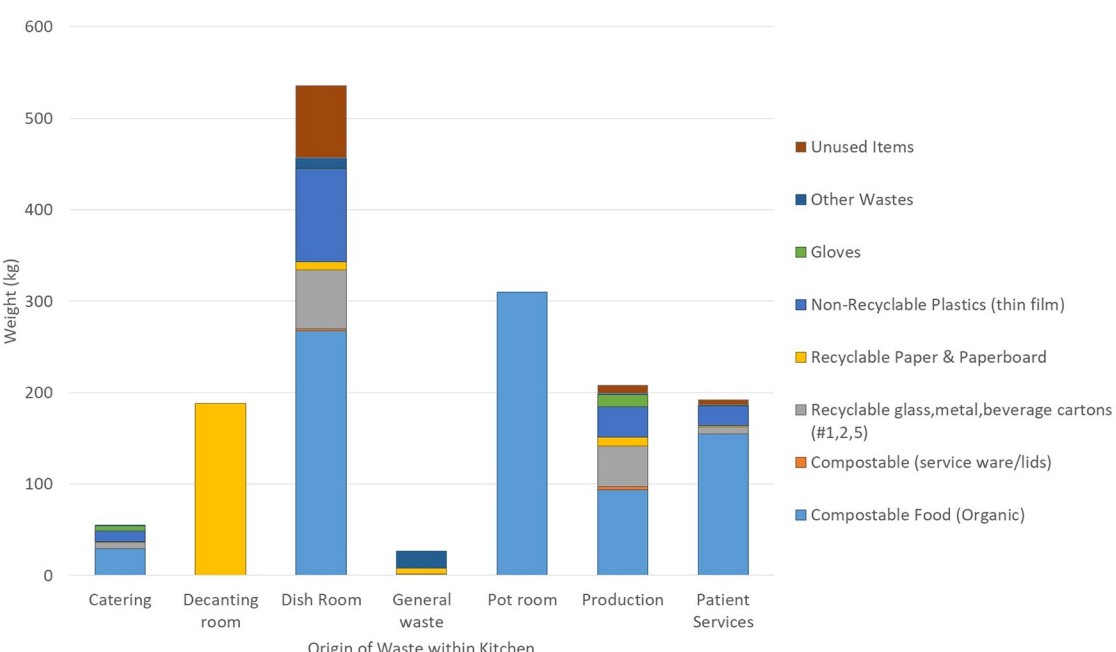

**Fig 1. Waste generation for case hospital kitchen on the day of the audit, with estimated compositions by kitchen zone.**
Estimated composition determined by extrapolation from detailed audits of 20% of waste bag sample. "Unused items" refers to unused food found in original unopened packaging. "Other wastes" refers to garbage that does not fit into other categories (e.g., disposable flatware).

dish room or decanting room (unlabeled garbage bags were accidentally used around 7pm, but the issue was quickly remedied).

Detailed audits were conducted on 34 bags (20% of daily total), representing 272.9 kg of waste (18%). The only kitchen areas without detailed audits were the decanting room and the pot room. In observations prior to and on the day of the audit, the decanting room generated mainly cardboard waste and some plastic shrink wrap (the total weight considered to be recycling) and the pot room generated only compostable waste (food liquid and solids). Extrapolating the detailed audits to the daily total by kitchen area reveals the potential amount of recyclable and compostable material originating in each kitchen area (Fig 1). Recyclable paper products made up 215.8 kg or 14% of the total waste. Of this, 93% is already being captured for recycling. Recyclable metals, plastics, or glass made up 124.8 kg (8%) of total waste, of which 20% is already being captured for recycling. Organic materials that could be sent to a composting facility made up 861.6 kg (57%) of the total waste with compostable service ware (lids, spoons, and forks) making up 5.2 kg of total waste (<1% of total and about 1% of compostable waste). Thin film plastics made up 168.1 kg (11%) and gloves represented 20.5 kg (1%). Unused items, or unused food found in unopened original packaging including drink cartons and individually wrapped food items, made up 93.5 kg or 6% of total waste. Most unused items came from the dish room, where returned patient trays are washed, although some originated in production, where expired and near-expired items from the freezers and refrigerators are disposed per regulations.

Waste generation and type of waste generated varied throughout the day, with more recyclable materials emerging early in the day, shown in Fig 2. Peaks in waste generation occurred in late morning before lunch time and mid-afternoon before dinner. An additional peak occurred at the end of the day, when the freezers and refrigerators were cleaned, resulting in

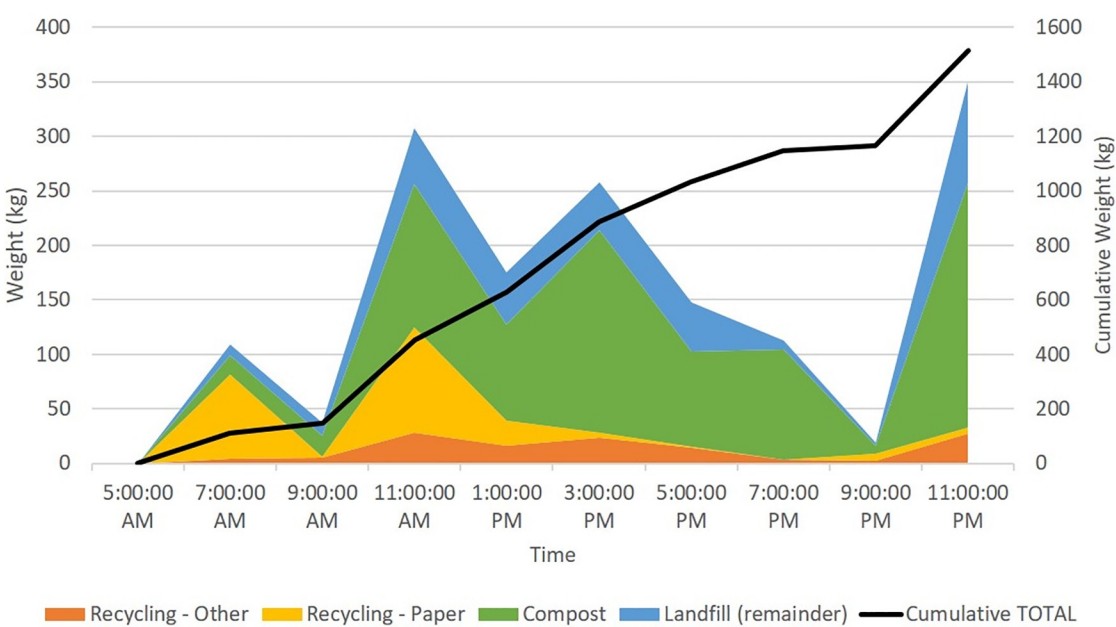

**Fig 2. Estimated composition of waste generation of case study hospital on day of audit over time.**

the disposal of expired supplies. Amount of compostable material ranged between 16% of total waste (morning) to 89% of total waste (around 7PM), though these factors may vary depending on the day (e.g., a weekend instead of a weekday).

Extrapolating the detailed audit to all day audit results show some large potential for waste diversion, though there are operational and regulatory barriers to implementing diversion strategies. In the current or Base Case scenario, the only waste diverted from landfill by the kitchen consisted of 200.7 kg of recyclable cardboard and 24.9 kg of metal, plastic, or glass (approximately 15% of total waste daily), shown in Fig 3. In an ideal scenario, all possible compostable food and service ware items and all recyclable items would be captured and diverted. If the hospital's daily measured generation of 861.6 kg of compost, 215.8 kg of paper, and 124.8 kg of other recycling were to be properly diverted, the total waste to landfill could be reduced by 76% by weight (Fig 3), resulting in only 313.0 kg of total waste to landfill daily or 91,330 kg of waste to landfill annually. For this kitchen, a more realistic scenario limits composting to catering, production, patient services, and the pot room, for an estimated 55% reduction in waste to landfill (including 588.5 kg daily to compost), resulting in 586.1 kg of total waste to landfill daily or 171,002 kg of waste to landfill annually. However, even this scenario many be overestimated due to potential barriers associated with logistics, space constraints, and resources required.

Based on these waste diversion scenarios, we estimated that the hospital kitchen generates 294,466 kg $CO_2$e per year from its current waste disposal practices (Base Case), in which all cardboard is recycled, some metals and plastics are recycled, and all other waste is sent to landfill (Fig 4). This is equivalent to the annual emissions from 64 standard US passenger cars and would require 386 acres of US forest growing for one year to absorb the released greenhouse gases [17]. Implementing the "ideal scenario" of diverting all possible materials would result in cumulative carbon emissions of 29,913 kg $CO_2$e annually (a 90% reduction in GHGs), equivalent to emissions from just less than 7 passenger vehicles on the road each year. This hospital

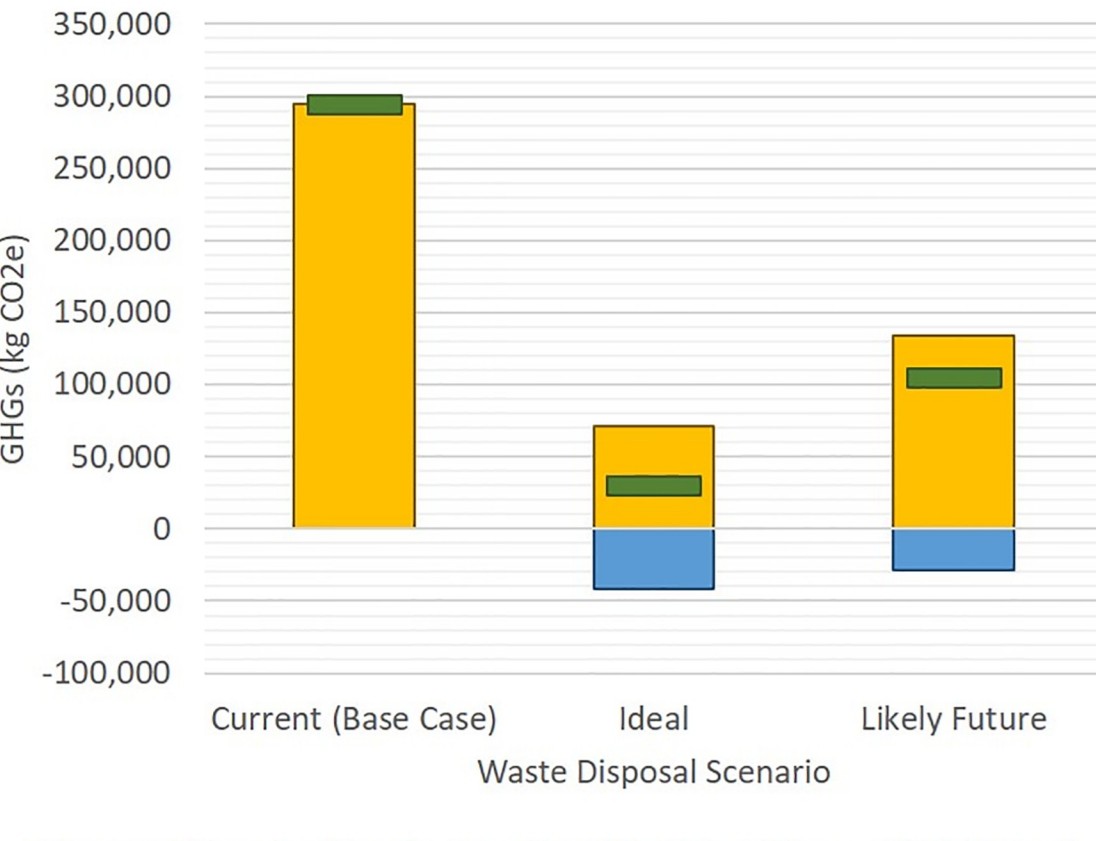

**Fig 3. Estimated daily waste generation and diversion scenarios for case hospital kitchen, based on one-day audit.** Current (Base Case) = kitchen diverts cardboard and some metal recycling only; Ideal = recycling and composting of all possible materials; Likely Future = recycling and composting of likely materials based on feedback from kitchen staff.

kitchen's likely waste diversion scenario would result in emissions of 105,441 kg $CO_2$e per year (a 64% reduction from baseline), equivalent to 23 passenger vehicles on the road each year.

## Discussion

In this study, we found that a single hospital in NYC annually generates 442,067 kg of waste (487 US tons) and emits approximately 294,466 kg of $CO_2$e from waste disposal. The majority of this waste (85%, 376,247 kg or 415 US tons annually) is currently sent to landfill. If the hospital were to increase recycling rates and implement composting in a feasible manner, it could reduce landfilled waste by 205,245 kg (226 US tons, or 55% reduction) and GHG emissions by 189,025 kg $CO_2$e (64% reduction). Although an ideal waste diversion scenario would divert 76% of waste from landfill and reduce GHG emissions by 90%, it would be particularly challenging to achieve due to available commercial composting facilities, vendor restrictions, staffing resources, space constraints, logistics, and expenses. Despite a dearth of data on waste diversion in other hospitals, one case study of a 250-bed hospital in San Francisco estimated that an institutional waste diversion program could result in an annual reduction of 51 US tons of landfilled waste [18]. That estimate is lower than the one found in this study, which is likely due to differences in the size of the hospitals' patient populations and foodservice operations, and variations in waste diversion strategy design.

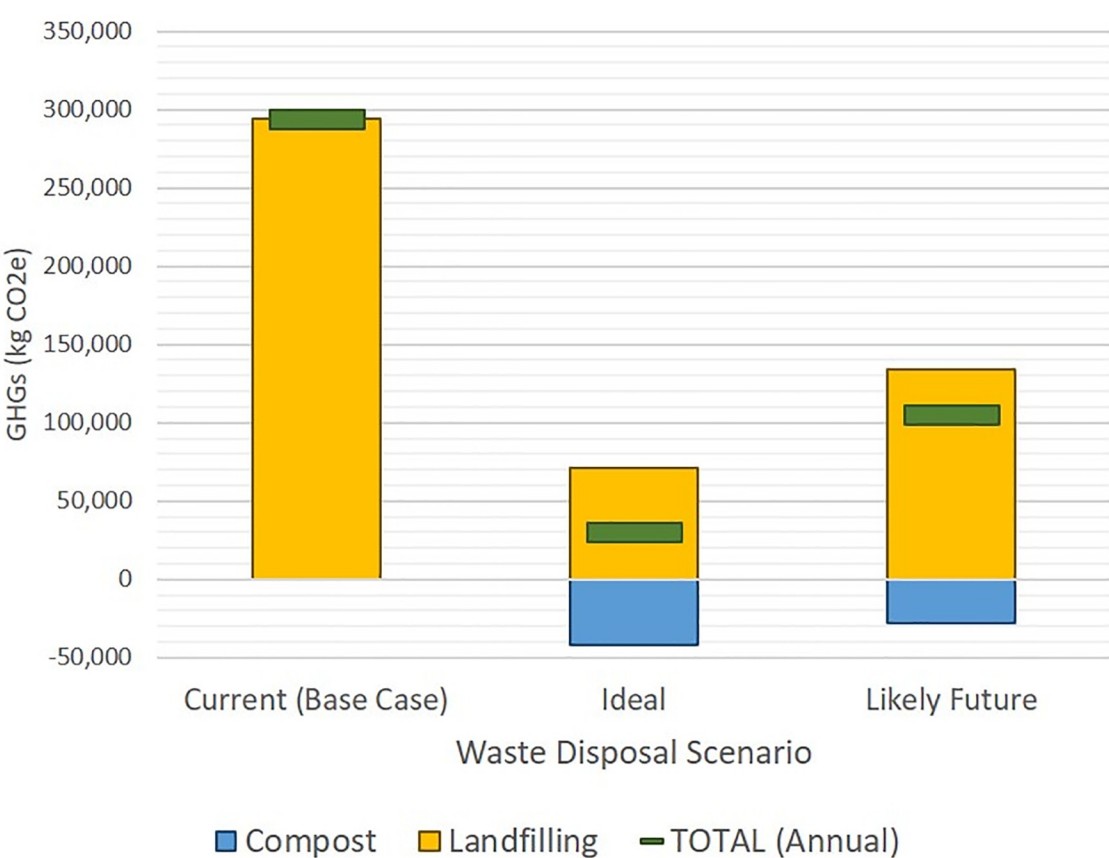

**Fig 4. Estimated annual carbon emissions from waste diversion scenarios in the case hospital kitchen.** Current (Base Case) = kitchen diverts cardboard and some metal recycling only; Ideal = recycling and composting of all possible materials; Likely Future = recycling and composting of likely materials based on feedback from kitchen staff. Greenhouse Gases (GHGs) from composting are assumed to be carbon negative. No carbon credit is issued for recycling, so it is not included in this figure.

Despite structural differences between hospitals, diverting organic and recyclable hospital waste from landfills could be highly impactful if implemented across NYC, as the city is home to 41 hospitals and 15,841 beds [19]. An extrapolation of our findings across all NYC hospitals based on the number of beds suggests that composting in NYC hospitals could result in a 3.9 million-kg (4,200 US tons) reduction in landfilled waste and 3.5 million-kg (3,900 tons) reduction in GHGs, the equivalent GHG savings as removing 1,674 passenger vehicles from the road each year [17]. Based on these data and existing literature, several strategies and policy changes may help (1) prevent food waste and (2) divert remaining wastes in institutional food services.

First, hospitals can focus on operational procedures to decrease food waste and emissions, as our study found that 35% of food waste came from the area that cleans inpatient food trays, and 7% of food was discarded unopened. Studies suggested that creating greater patient autonomy in food decisions can reduce tray waste [20]. Some hospitals allow patients to select their meal from a menu, while still incorporating their recommended dietary restrictions, and allow patients to select which condiments, beverages, and utensils they prefer. This is done at NYU Langone Health. However, state and federal regulations often lead hospital food services and other institutions to deliver items that patients will ultimately waste, such as milk cartons [21, 22]. Other studies of institutional food waste have identified additional strategies to mitigate

tray or plate waste, including: smaller serving sizes and educational campaigns [20, 23–25], implementing support systems to overcome clinical barriers [26], and addressing service issues such as complex ordering systems or environmental factors such as inappropriate meal times or unpleasant ward surroundings [12, 27]. Some issues, such as serving sizes, are controlled by dietary regulation and USDA guidelines making it more difficult for hospitals to adjust practices that lead to waste. Strategies that are not restricted by regulation may be easier to implement based on the size and structure of the hospital and its food service department. Regular and ongoing tracking of food waste is recommended as part of a reduction strategy; however, it is not often performed in hospital food services [28, 29]. Tracking food waste can help establish a baseline, identify opportunities for reduction, and report progress to staff and leadership.

Second, our study found that the production area of the kitchen generated 12% of total waste, suggesting that upstream strategies focusing on procurement, inventory management practices, and production approaches may be equally helpful in minimizing potential food waste and increasing opportunities for diversion. Careful planning to avoid unnecessary food waste is core to any food service, as this will also save money. Though not measured in this study, procurement practices can reduce upstream emissions. These include prioritizing locally grown, organic, and in-season produce (which notably may not always reduce GHGs); reducing animal proteins in meals; offering more vegetarian and vegan options; sourcing produce and meats more sustainably; and purchasing reusable, compostable, or recyclable products [29–32]. Operational improvements such as energy and water efficiency upgrades can also reduce the larger footprint of food services [33, 34].

Third, if food is still usable, preference should be given to ensuring the food ends up being eaten. However, one concern often cited is the legal ramifications of donating unused food. Policymakers and hospitals could become familiarized with the food waste policies in their states. Some US states, for example, protect donors from legal consequences if someone gets sick as a result of consuming donated food. New York State recently passed a Food Donation and Food Scraps Recycling Law, which will require large generators of food scraps to donate excess edible food and recycle all remaining food scraps starting January 1, 2022; however, New York City is exempt as it has its own policy and hospitals are also exempt [35]. Unfortunately, logistical concerns for implementing a donation program can often be prohibitive for many hospitals, though these issues may present an opportunity for aspiring businesses or non-profits.

Finally, food and recyclable wastes should be diverted from landfills, either through composting, recycling, or anaerobic digestion, which uses microorganisms to break down organic material into biogas, which can be used as a renewable energy source. Substantial barriers exist to implementing these strategies, however. Many hospitals need to establish contracts with private or commercial composters/anaerobic digestion operators, and for many, recycling is a much more commonly implemented waste diversion initiative [36]. Cities might consider supporting local composting infrastructure, and enacting legislation that encourages large institutions to utilize these waste streams. For example, prior to COVID-19, the NYC Department of Sanitation (DSNY) offered composting in residential areas and for non-profit organizations. DSNY also partners with a non-profit called GrowNYC that facilitates food scrap drop-off sites in local farmers markets.

These recommendations may be applied to other institutional kitchens, such as cafeterias at large universities or military bases. However, more research is needed to understand how food waste can be reduced in other sectors responsible for feeding thousands of people. Given the complex factors that may be involved in reducing food waste (e.g., changing the status quo, composting markets, and local regulations), future studies should also qualitatively examine

perceived barriers to reducing and diverting food waste in hospitals and other large food service providers [32, 34].

## Limitations

This study analyzed only one hospital kitchen on a single day. Variability obviously exists by day of the week, season, and year, and by the way in which a hospital's food service operates. Additionally, this study did not capture staff and visitor waste from retail meals, as the cafeteria's front-facing waste stream does not come through the kitchen. As a result, this study underestimates the hospital's total food service wastes and potential diversion opportunities. In addition, this study used a sampling frame to estimate the composition of daily kitchen wastes. This study also utilized GHG factors from the US EPA's WARM model, which contains its own assumptions about landfilling, recycling, and composting. Other GHG models may estimate emissions differently. Future studies should audit waste in detail over a longer time period (e.g., multiple days throughout the year), and further investigate opportunities to reduce staff and visitor cafeteria waste. Future studies should also investigate other opportunities for organic waste diversion, including anaerobic digestion, and calculate and compare costs of various waste diversion strategies, including composting, anaerobic digestion, and recycling.

## Conclusion

Hospital kitchens and other institutional food services generate a substantial amount of solid waste and greenhouse gas emissions. However, there are ample opportunities to reduce and divert this waste from methane-producing landfills. Using waste audit methods at a case location, we found that the hospital kitchen in this study annually sends 85% of its waste—415 US tons—to landfill, but that it could feasibly reduce its waste to landfill by 55% and subsequently reduce its GHG emissions by 64%. This study provides a valuable framework from which hospitals and policymakers in NYC and elsewhere can begin to measure and subsequently reduce institutional food waste. It is also particularly timely, as NYC's OneNYC strategic plan has targeted sustainability goals including Zero Waste by 2030 and carbon neutrality by 2050.

## Supporting information

**S1 Data.**
(XLSX)

## Acknowledgments

The researchers would like to thank Purnima Prasad, Omni Cassidy, Tenay Greene, and Josh Arshonsky for their help conducting the waste audits. We would also like to thank NYU Langone Health Department of Food & Nutrition Services and NYU Langone Health Department of Building Services workers for their assistance in making the audit a success.

## Author Contributions

**Conceptualization:** Cassandra L. Thiel, Jenna Agins, Jocelyn Gan, Jeffrey Held.

**Data curation:** Cassandra L. Thiel, SiWoon Park, Jenna Agins, Jocelyn Gan, Marie A. Bragg.

**Formal analysis:** Cassandra L. Thiel, SiWoon Park, Jocelyn Gan.

**Investigation:** Cassandra L. Thiel, SiWoon Park, Jocelyn Gan, Jeffrey Held, Marie A. Bragg.

**Methodology:** Cassandra L. Thiel, SiWoon Park, Jenna Agins, Jocelyn Gan, Jeffrey Held, Marie A. Bragg.

**Project administration:** Cassandra L. Thiel, Jenna Agins, Jocelyn Gan, Jeffrey Held, Amy Horrocks.

**Resources:** Cassandra L. Thiel, Jocelyn Gan, Jeffrey Held, Amy Horrocks, Marie A. Bragg.

**Software:** Cassandra L. Thiel.

**Supervision:** Cassandra L. Thiel, Amy Horrocks, Marie A. Bragg.

**Validation:** Cassandra L. Thiel, Aviva A. Musicus, Jocelyn Gan, Jeffrey Held.

**Visualization:** Cassandra L. Thiel, SiWoon Park, Jocelyn Gan.

**Writing – original draft:** Cassandra L. Thiel, SiWoon Park, Aviva A. Musicus, Amy Horrocks, Marie A. Bragg.

**Writing – review & editing:** Cassandra L. Thiel, Aviva A. Musicus, Jenna Agins, Jocelyn Gan, Jeffrey Held, Amy Horrocks, Marie A. Bragg.

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
