## [Decision Letter · Decision Letter 0]

15 Dec 2020

PONE-D-20-30816

Waste generation and carbon emissions of a hospital kitchen in the US: potential for waste diversion and carbon reductions

PLOS ONE

Dear Dr. Cassandra L. Thiel,

Thank you for submitting your manuscript to PLOS ONE. After careful consideration, we feel that it has merit but does not fully meet PLOS ONE’s publication criteria as it currently stands. Therefore, we invite you to submit a revised version of the manuscript that addresses the points raised during the review process.

Please submit your revised manuscript by 10 January,2021. If you will need more time than this to complete your revisions, please reply to this message or contact the journal office at plosone@plos.org. Please include the following items when submitting your revised manuscript:

We look forward to receiving your revised manuscript.

Kind regards,

Balasubramani Ravindran, Ph.D

Academic Editor

PLOS ONE

Journal Requirements:

2. We note that you have stated that the data underlying the results will be available from the corresponding author. When possible, we recommend authors deposit restricted data to a repository that allows for controlled data access. If this is not possible, directing data requests to a non-author institutional point of contact, such as a data access or ethics committee, helps guarantee long term stability and availability of data. Providing interested researchers with a durable point of contact ensures data will be accessible even if an author changes email addresses, institutions, or becomes unavailable to answer requests.  As such, we ask that you amend your Data Availability Statement to provide a durable point of contact for data requests.

For more information on unacceptable data access restrictions, please see http://journals.plos.org/plosone/s/data-availability#loc-unacceptable-data-access-restrictions

Reviewers' comments:

Reviewer's Responses to Questions

**Comments to the Author**

1. Is the manuscript technically sound, and do the data support the conclusions?

Reviewer #1: Yes

Reviewer #2: No

Reviewer #3: Partly

2. Has the statistical analysis been performed appropriately and rigorously? 

Reviewer #1: N/A

Reviewer #2: No

Reviewer #3: Yes

3. Have the authors made all data underlying the findings in their manuscript fully available?

Reviewer #1: Yes

Reviewer #2: Yes

Reviewer #3: Yes

4. Is the manuscript presented in an intelligible fashion and written in standard English?

Reviewer #1: Yes

Reviewer #2: Yes

Reviewer #3: Yes

5. Review Comments to the Author

Reviewer #1: The manuscript technically sound and authors made a good attempt to quantize the food waste generated in a hospital kitchen of US and will be very helpful for the society to take actions to reduce the amount of food waste and utilize the generated food waste for energy production and composting purposes which will also help to reduce GHG emission. As author already mentioned that the study is based on one day audit, if it will be elaborated for long time with more results, i.e., characterization of food waste for its utilization for energy generation, the research will be more fruitful for the society.

I want to suggest some minor corrections which will help the author to make the manuscript more fit to publish in PLOS ONE:

1. 75% + should be written as 'more than 75%' (line 87) and 90%+ should be written as 'more than 90%' (line 88).

2. Are terms 'all day audit' (line 143) and 'detailed audits' (line 145) given by author? or if reported before than reference should be added.

Reviewer #2: 1. Introduction could be more concise to make it easy for understanding. A lack of flow between the sentences are observed. Too much literature cited but lack of deep discussion observed.

2. Research highlights could be added that include main findings instead of general statements.

3. Please update your work by comparing the latest articles published in journals.

4. In scientific publication reference of website is not acceptable e.g US EPA. Greenhouse Gas Equivalencies Calculator 2018.

5. Some of the references seems incomplete like Mirosa M, Munro H, Mangan-Walker E, Pearson D. Reducing waste of food left on plates. British Food Journal. 2016. There are more

6. Reseult lacks statistical analysis

7. Limitations itself says the story of data, as Variability obviously exists by day of the week, season, and year, and by the way in which a hospital’s food service operate. Such study requires long term study. Study is very concise.

Reviewer #3: The manuscript titled " Waste generation and carbon emissions of a hospital kitchen in the US: potential for waste diversion and carbon reductions" by Thiel et al., did a survey of waste generation from NYC hospital from New York, categorized the biodegradable and recyclable fractions, quantified and calculated the GHG emission contributions by its current disposal practices. The study also proposed few alternate options to recycle the waste as compost and reduce the burden of landfill operation in future. The study is very basic and concluded based on one singe data set, which is not realistic. The composition and quantity of the waste might not be the same throughout the year, while the study doesn't include any scientific elements. Composting process also emit lot of GHGs (N2O and CO2) and this would be more than landfill based GHG emissions that is calculated in this study. Overall the paper required more detailed study and data collection, interpretations to be accepted for publication. Composting process is not ideal food waste, instead anaerobic digestion of food waste and energy recovery could be considered as a option. Cost and GHG emissions from the food waste processing by landfilling, composting and anaerobic digestion need to be compared and discussed.

6. PLOS authors have the option to publish the peer review history of their article (what does this mean?). If published, this will include your full peer review and any attached files.

Reviewer #1: No

Reviewer #2: No

Reviewer #3: No

---

## [Author Response · Author response to Decision Letter 0]

13 Jan 2021

Responses to all reviewer and editor comments can be found in the word document attached to this submission.

---

## [Decision Letter · Decision Letter 1]

10 Feb 2021

Waste generation and carbon emissions of a hospital kitchen in the US: potential for waste diversion and carbon reductions

PONE-D-20-30816R1

Dear Dr. Cassandra L. Thiel,

We’re pleased to inform you that your manuscript has been judged scientifically suitable for publication and will be formally accepted for publication once it meets all outstanding technical requirements.

Kind regards,

Balasubramani Ravindran, Ph.D

Academic Editor

PLOS ONE

Reviewers' comments:

Reviewer's Responses to Questions

**Comments to the Author**

1. If the authors have adequately addressed your comments raised in a previous round of review and you feel that this manuscript is now acceptable for publication, you may indicate that here to bypass the “Comments to the Author” section, enter your conflict of interest statement in the “Confidential to Editor” section, and submit your "Accept" recommendation.

Reviewer #1: All comments have been addressed

Reviewer #3: All comments have been addressed

2. Is the manuscript technically sound, and do the data support the conclusions?

Reviewer #1: Partly

Reviewer #3: Yes

3. Has the statistical analysis been performed appropriately and rigorously? 

Reviewer #1: No

Reviewer #3: Yes

4. Have the authors made all data underlying the findings in their manuscript fully available?

Reviewer #1: Yes

Reviewer #3: Yes

5. Is the manuscript presented in an intelligible fashion and written in standard English?

Reviewer #1: Yes

Reviewer #3: Yes

6. Review Comments to the Author

Reviewer #1: Authors have incorporated all the suggested corrections in a very well manner. I wish for the best and suggest that this study must be elaborated with more techniques and results to enhance the fruitfulness of the process. Authors can also make an attempt to produce value added products from kitchen waste via anaerobic digestion process.

Reviewer #3: Authors addressed all the questions and revised the manuscript. Now it can accepted for publication.

7. PLOS authors have the option to publish the peer review history of their article (what does this mean?). If published, this will include your full peer review and any attached files.

Reviewer #1: No

Reviewer #3: No

---

## [Editor Report · Acceptance letter]

19 Feb 2021

PONE-D-20-30816R1 

Waste generation and carbon emissions of a hospital kitchen in the US: Potential for waste diversion and carbon reductions 

Dear Dr. Thiel:

I'm pleased to inform you that your manuscript has been deemed suitable for publication in PLOS ONE. Congratulations! Your manuscript is now with our production department. 

Kind regards, 

on behalf of

Dr. Balasubramani Ravindran 

Academic Editor

PLOS ONE